# Playing with Transformer at 30+ FPS via Next-Frame Diffusion

## Abstract

Autoregressive video models offer distinct advantages over bidirectional diffusion models in creating interactive video content and supporting streaming applications with arbitrary duration. Nonetheless, achieving real-time video generation remains a significant challenge for such models, primarily due to the high computational cost associated with diffusion sampling and the hardware inefficiencies inherent to autoregressive generation. In this work, we present Next-Frame Diffusion (NFD), an autoregressive diffusion transformer that incorporates block-wise causal attention, enabling iterative sampling and efficient inference via parallel token generation within each frame. To address the aforementioned challenges, we introduce two innovations: (1) We for the first time extend consistency distillation to the video domain and adapt it specifically for video models, enabling efficient inference with few sampling steps; (2) To fully leverage parallel computation, motivated by the observation that adjacent frames often share the identical action input, we propose speculative sampling. In this approach, the model generates next few frames using current action input, and discard speculatively generated frames if the input action differs. Experiments on a large-scale action-conditioned video generation benchmark demonstrate that NFD beats autoregressive baselines in terms of both visual quality and sampling efficiency. We, for the first time, achieves autoregressive video generation at over 30 Frames Per Second (FPS) on an A100 GPU using a 310M model.

## 1 Introduction

Diffusion models have shown remarkable success across diverse generative tasks (Ho et al., 2020; Dhariwal & Nichol, 2021; Arriola et al., 2025; Xiang et al., 2025; Chi et al., 2023), offering strong performance in terms of both visual quality and diversity. In video generation, substantial progress has been driven by Diffusion Transformers (DiTs) (Peebles & Xie, 2023), which utilize bidirectional attention across all frames to capture complex spatio-temporal dependencies (OpenAI, 2024; Polyak et al., 2024). However, generating all frames in parallel constrains their applicability to interactive and streaming scenarios. In contrast, autoregressive video models (Kondratyuk et al., 2024; Guo et al., 2025), which build upon successes in language modeling (Brown et al., 2020; Touvron et al., 2023), support temporally causal generation but often struggle to maintain high visual fidelity due to vector quantization (Esser et al., 2021).

A promising solution for interactive and streaming video generation is to combine the high-fidelity, continuous-space capabilities of diffusion models with the temporal causality and controllability of autoregressive models. Despite its potential, achieving real-time generation presents three major challenges: (1) token-level sequential generation incurs substantial latency as spatial and temporal resolution increase, creating a bottleneck for real-time applications; (2) although diffusion (Sohl-Dickstein et al., 2015; Ho et al., 2020; Song et al., 2020) and flow-matching (Lipman et al., 2023; Liu et al., 2023) models can produce high-quality outputs, they typically require tens to hundreds of network evaluations during sampling, resulting in significant computational overhead; (3) Transformer-based sampling is often bandwidth-bound (Shazeer, 2019), necessitating careful exploitation of parallel computation to achieve efficiency.

To address these challenges, recent research has investigated approaches to enable parallel generation within autoregressive diffusion models, leveraging both U-Net (Chen et al., 2024a; Kim et al., 2025;

Valevski et al., 2024) and Transformer (Yin et al., 2024c; Gu et al., 2025) architectures. For example, CausVid (Yin et al., 2024c) adapts a bidirectional diffusion transformer into an autoregressive framework. Furthermore, to accelerate generation, it employs Distribution Matching Distillation (DMD) (Yin et al., 2024b;a) to distill the diffusion process into a four-step generator. While effective, this approach remains constrained by its reliance on a bidirectional teacher and does not fully exploit the potential for parallel computation.

In this work, we thereby introduce Next-Frame Diffusion (NFD), a diffusion-based video transformer tailored for efficient and high-fidelity autoregressive video generation to overcome these challenges. (1) To achieve scalability and efficiency, NFD adopts a block-wise causal attention mechanism, which enables bidirectional self-attention within individual frames while ensuring that each frame is conditioned only on past frames. (2) To mitigate the latency from large diffusion sampling steps, we extend sCM (Lu & Song, 2024; Chen et al., 2025) to the video domain, enabling fast inference by reducing the number of function evaluations (NFE) to just a few steps without compromising output quality. (3) To further leverage parallel computation, we draw on the empirical observation that adjacent frames frequently share the identical action input. Based on this, we propose speculative sampling, which pre-generates the next few frames by conditioning the model on the current action. If a change in the action input is subsequently detected, the speculatively generated frames are discarded, and new frames are generated to align the updated action. Moreover, to mitigate error accumulation inherent in autoregressive generation, we corrupt context frames by adding a small amount of Gaussian noise to generated frames during sampling (Chen et al., 2024a; Valevski et al., 2024).

We train Next-Frame Diffusion (NFD) on a large-scale action-conditioned video generation benchmark (Baker et al., 2022; Guo et al., 2025), consisting of paired gameplay videos and corresponding action sequences. Empirical results demonstrate that NFD outperforms existing autoregressive baselines, achieving real-time autoregressive video generation and improved visual fidelity.

In summary, our contributions are threefold: (1) Building on our pretrained block-causal video Diffusion Transformer, we propose NFD as the first framework to incorporate continuous-time Consistency Models into video generation, thereby mitigating discretization errors inherent in prior discretized distillation approaches (Yin et al., 2024c). (2) We propose speculative sampling, marking the first application of this technique to accelerate inference in video diffusion models. (3) By combining block-causal diffusion distillation with speculative sampling, NFD achieves, for the first time, real-time autoregressive video generation at over 30 Frames Per Second (FPS) on a single NVIDIA A100 GPU with a 310M-parameter model.

## 2 RELATED WORKS

**Autoregressive Video Generation.** Autoregressive models are naturally suited to streaming and interactive settings due to their causal structure. Recent works like VideoPoet (Kondratyuk et al., 2024), iVideoGPT (Wu et al., 2024), and MineWorld (Guo et al., 2025) applied large language modeling strategies to video generation, modeling frame sequences through sequential token prediction. However, these approaches typically rely on vector quantized representations (Esser et al., 2021), which can compromise visual fidelity. Moreover, while effective in capturing temporal dynamics, these models suffer from inefficiencies in inference due to their token-by-token sampling. Our work shares with these models the autoregressive structure, but improves both fidelity and sampling speed by operating in continuous space and leveraging diffusion-based sampling with parallelism. Previous work have explored parallel next-frame prediction (Chen et al., 2024a; Gu et al., 2025). Diffusion Forcing introduces varying noise levels across frames (Chen et al., 2024a), while CausVid distills a bidirectional teacher model into an causal student (Yin et al., 2024c). Building on these successes, we present Next-Frame Diffusion (NFD) in an action-conditioned gaming environment, and further introduce continuous-time Consistency Models and speculative sampling to achieve real-time interactive generation.

**World Models.** World models (Ha & Schmidhuber, 2018) have demonstrated significant potential in training reinforcement learning agents across diverse environments (Schrittwieser et al., 2020; Hafner et al., 2023). In the context of autonomous driving, several studies have been proposed to predict multiple plausible future trajectories conditioned on various prompts (Hu et al., 2023; Russell et al., 2025; Zheng et al., 2024; Gao et al., 2024; Zhao et al., 2025). In real-world robotics

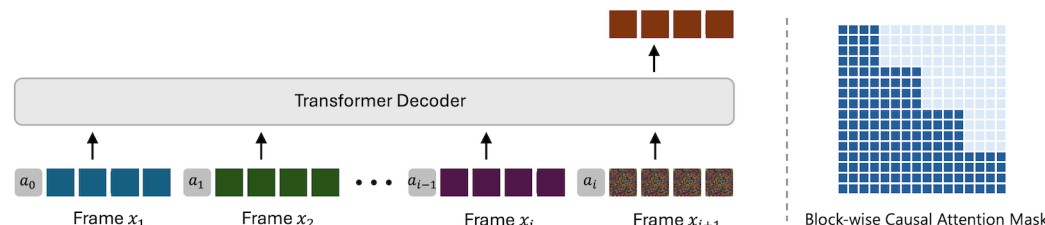

Figure 1: We present Next-Frame Diffusion (NFD), an autoregressive diffusion transformer that employs block-wise causal attention. This design enables parallel generation of multiple tokens for an entire frame, thereby enhancing sampling efficiency and better aligning with hardware constraints.

and embodiment learning, recent researches like UniPi (Du et al., 2023) or UniSim (Yang et al., 2024a) leveraged generative modeling by reformulating the decision-making process as a conditional generation task, conditioned on inputs such as textual descriptions (Du et al., 2023; Yang et al., 2024a; Zhou et al., 2024; Xiang et al., 2024; Agarwal et al., 2025) or latent action representations (Chen et al., 2024b; Bu et al., 2025). For gaming, some important works simulated interactive video games with neural networks (Ha & Schmidhuber, 2018; Kim et al., 2020; Bruce et al., 2024; Valevski et al., 2024; Alonso et al., 2024; Guo et al., 2025). Nevertheless, most of these approaches are limited to simulation speed or visual quality. Several methods (Bruce et al., 2024; Valevski et al., 2024; Alonso et al., 2024) have also achieved next-frame prediction through diffusion-based approaches using U-Net backbone (Ronneberger et al., 2015). In contrast, our method leverages a transformer-based architecture with block-wise causal attention, with advantages in both scalability and efficiency.

**Efficient Diffusion Models.** Despite their impressive generative quality, diffusion models typically incur a high computational cost during sampling. A growing body of work focuses on improving sampling efficiency, either through efficient solvers (Lu et al., 2022; Song et al., 2020), speculative decoding (De Bortoli et al., 2025; Christopher et al., 2024), sparsity (Cheng et al., 2025; Wang et al., 2024), or distillation (Salimans & Ho, 2022; Song et al., 2023; Lu & Song, 2024; Yin et al., 2024b). For example, Song et al. firstly introduced the Consistency Model (CM), which exploits the self-consistency property inherent in generative ordinary differential equations (ODEs) used in diffusion models. By minimizing the discrepancy in the self-consistency function, their approach enables more efficient training. Consequently, Chen et al. (Chen et al., 2025) transformed the pre-trained flow matching models into TrigFlow (Lu & Song, 2024), and accelerated consitency models via hybrid distillation, enabling sampling with 1-4 steps. Building upon these advancements in the image domain, our model adapts and extends their effectiveness to the video generation setting by introducing several key modifications. These improvements enable efficient sampling in a few steps while preserving high visual fidelity.

## 3 NEXT-FRAME DIFFUSION

We introduce the proposed Next-Frame Diffusion (NFD) framework in this section, and techniques of improving the sampling efficiency of NFD in the next section. We start with the problem definition.

**Problem Definition.** We focus on the action-conditioned video generation task (Guo et al., 2025; Alonso et al., 2024) to develop and validate our method. Let $x_i$ denote the $i$-th video frame and $a_i$ the corresponding user action taken upon observing $x_i$ to obtain the next frame $x_{i+1}$. The model is conditioned on the sequence of past frames $\{x_{1:i}\}$ and action $a_i$, and is trained to predict the next frame $x_{i+1}$. We illustrate the overall architecture in Fig. 1. Block-wise causal attention is adopted to achieve bidirectional self-attentions among patches within each individual frame while preserving causal dependencies across frames.

### 3.1 ARCHITECTURE

The architecture of NFD contains a tokenizer that transforms raw visual signals to latent representations, and a Diffusion Transformer (DiT) (Peebles & Xie, 2023) that generates these latents. We introduce some key components in this section.

**Tokenizer.** To enable the frame-level interaction with the model, we employ an image-level tokenizer (Li et al., 2024) to transform each frame into a sequence of latent representations. For actions, we follow previous works (Baker et al., 2022; Guo et al., 2025) to quantize camera angles into discrete bins, and categorize other actions into 7 exclusive classes, each represented by a unique token (Guo et al., 2025).

**Block-wise Causal Attention.** We propose a Block-wise Causal Attention mechanism that combines bidirectional attention within each frame and causal dependencies across frames to model spatio-temporal dependencies efficiently. Specifically, for each token in a frame, it will attend to all tokens within the same frame (i.e., intra-frame attention), as well as to all tokens in preceding frames (i.e., causal inter-frame attention). In contrast to the computationally intensive 3D full attention (OpenAI, 2024; Ho et al., 2022; Kong et al., 2024; Yang et al., 2024b), our approach reduces the overall cost by $50\%$ (see Fig. 1), enabling hardware-efficient and streaming prediction of all tokens in the next frame in parallel.

**Action Conditioning.** We utilize a linear layer to map the actions into action vectors and explore various DiT architectural designs to incorporate action conditioning into the model. Following the approach of DiT (Peebles & Xie, 2023), we investigate three conditioning mechanisms independently: adaLN-zero blocks, cross-attention blocks, and in-context conditioning. We adopt adaLN-zero conditioning as it produces best performance empirically.

**3D Positional Embedding.** Following HunyuanVideo (Kong et al., 2024), we separate the head dimension of the query and key tokens into $[n_T, n_H, n_W]$, encoding their temporal and spatial correspondence independently. Specifically, we compute rotary frequency embeddings for each axis separately and concatenate them along the last dimension.

## 3.2 TRAINING AND SAMPLING

**Training.** We formulate our training pipeline based on Flow Matching (Lipman et al., 2023; Liu et al., 2023), aiming for both simplicity and stability. Given a video frame $x_i$, we assign an independent timestep $t$ and generate a noised version via linear interpolation:

$$x_i^t = (1 - t)x_i^0 + t\epsilon, \quad \text{where} \quad \epsilon \sim \mathcal{N}(0, I). \tag{1}$$

This allows us to define a target velocity vector pointing from the clean frame $x_i^0$ toward the noise $\epsilon$, given analytically as $v_i^t \equiv \epsilon - x_i^0$. To prioritize learning over intermediate timesteps, we adopt the timestep sampling strategy from SD3 (Esser et al., 2024) and sample $t \sim \sigma(\mathcal{N}(0, 1))$. Conditioned on the autoregressive context of preceding frames $\{x_j\}_{j<i}$ and action $a_{i-1}$, the model predicts the velocity given the noised frame $x_i^{t_i}$ and its timestep $t_i$. Training minimizes the following Flow Matching loss:

$$\mathcal{L}_{\text{FM}} = \mathbb{E}_{x_i^0, \epsilon, t_i, a_{i-1}} \left[ \left\| v_\theta(x_i^{t_i} \mid \{x_{j<i}^{t_j}\}, t_i, a_{i-1}) - (\epsilon - x_i^0) \right\|_2^2 \right]. \tag{2}$$

**Sampling.** For sampling, we adopt DPM-Solver++ (Lu et al., 2022), a fast high-order ODE solver for efficient and accurate generation under flow-based models. At each decoding step, we reverse the noise interpolation process to reconstruct clean frames from their noised versions. Given the predicted velocity $v_\theta$, we recover the denoised frame $x_i^0$ with:

$$x_i^0 = \frac{x_i^{t_i} - t_i \cdot \epsilon_\theta}{1 - t_i}, \quad \text{where} \quad \epsilon_\theta = (1 - t_i) \cdot v_\theta + x_i^{t_i}.$$

This substitution leverages the learned velocity to approximate the noise component, enabling a deterministic reconstruction of clean frames from intermediate states.

## 4 ACCELERATED SAMPLING

While NFD enables parallel token sampling during inference, achieving real-time video generation remains challenging. This limitation is primarily due to the substantial computational overhead of diffusion-based sampling and the hardware inefficiencies associated with autoregressive generation processes. In this section, we introduce a set of methodological advancements aimed at improving the sampling efficiency of NFD, while preserving high visual fidelity in the generated video content.

## 4.1 CONSISTENCY DISTILLATION

Although DPM-Solver++ reduces the number of sampling steps to the order of tens, achieving real-time video generation remains challenging with tens of sampling steps. To further improve sampling efficiency, we extend consistency distillation (Lu & Song, 2024; Chen et al., 2025) to the video domain, and adapt it to the specific features of video data.

Specifically, the sCM framework (Lu & Song, 2024) leverages the TrigFlow model $F_\theta$ since it is a special case for Flow Matching and also aligns with EDM (Karras et al., 2022). Here we first get $F_\theta$ via:

$$F_\theta \left( \frac{x_i^{t_i'}}{\sigma_d}, t_i' \right) = \frac{1}{\sqrt{t_i^2 + (1-t_i)^2}} \left[ (1 - 2t_i)x_i^{t_i} + (1 - 2t_i + 2t_i^2)v_\theta(x_i^{t_i}, t_i) \right]. \tag{3}$$

For clarity, we denote $t_i$ as the timestep used in Flow Matching and $t_i'$ as the timestep used in TrigFlow. All omitted conditioning variables (e.g., $a_i$, $\{x_{j<i}^{t_j}\}$) are understood from context.

We compute the model input $x_i^{t_i}$ and timestep $t_i$ used in Flow Matching based on the TrigFlow timestep $t_i'$ and sample $x_i^{t_i'}$, using the following formulations:

$$x_i^{t_i} = \frac{x_i^{t_i'}}{\sigma_d} \cdot \sqrt{t_i^2 + (1-t_i)^2}, \quad t_i = \frac{\sin(t_i')}{\sin(t_i') + \cos(t_i')}.$$

Then the training objective of the sCM part becomes:

$$\mathcal{L}_{\text{sCM}} = \mathbb{E}_{x_i^{t_i'}, t_i'} \left[ \frac{e^{w_\phi(t_i')}}{D} \left\| F_\theta \left( \frac{x_i^{t_i'}}{\sigma_d}, t_i' \right) - F_{\theta^-} \left( \frac{x_i^{t_i'}}{\sigma_d}, t_i' \right) - \cos(t_i') \cdot \frac{\mathrm{d}f_{\theta^-}(x_i^{t_i'}, t_i')}{\mathrm{d}t_i'} \right\|_2^2 - w_\phi(t_i') \right]. \tag{4}$$

Here D denotes the dimension of $x_i$, $\theta^-$ denotes the `stopgrad` version of the model, and $f_\theta$ predicts the clean data by:

$$f_\theta(x_i^{t_i'}, t_i') = \cos(t_i') \cdot x_i^{t_i'} - \sin(t_i') \cdot \sigma_d \cdot F_\theta \left( \frac{x_i^{t_i'}}{\sigma_d}, t_i' \right), \tag{5}$$

Despite the success of existing methods in the image domain, these approaches remain insufficient for the challenges posed by video generation. To better adapt the optimization process to the video generation, we further introduce the following techniques:

**Independent Timestep for Each Frame.** TrigFlow operates over a time domain $t \in \left[0, \frac{\pi}{2}\right]$. For each frame $i$, we independently sample $\tan(t_i)$ from a log-normal proposal distribution defined by $e^{\sigma_d \tan(t_i)} \sim \mathcal{N}(P_{\text{mean}}, P_{\text{std}}^2)$. The parameters $P_{\text{mean}}$ and $P_{\text{std}}$ are shared across all frames and remain fixed throughout training.

**3D Tangent Normalization.** As discussed in sCM (Lu & Song, 2024), normalizing $\frac{\mathrm{d}f_{\theta^-}}{\mathrm{d}t}$ by $\left\| \frac{\mathrm{d}f_{\theta^-}}{\mathrm{d}t} \right\| + c$ reduces gradient variance during training. In our video setting, we use $\left\| \sum_i \frac{\mathrm{d}f_{\theta^-}}{\mathrm{d}t} \right\|$ as the normalization factor, where $i$ indexes video frames.

**Training.** To enhance generation quality, we introduce adversarial supervision with a frozen, pretrained teacher model $D$ equipped with discriminator heads. The adversarial loss is defined as:

$$\mathcal{L}_{\text{adv}} = \mathbb{E}_{x_i^0, s} \left[ \text{ReLU} \left( 1 - D(x_i^s, s) \right) \right] + \mathbb{E}_{x_i^0, s, t} \left[ \text{ReLU} \left( 1 + D(\hat{x}_i^s, s) \right) \right], \tag{6}$$

where $x_i^s$ and $\hat{x}_i^s$ are the noisy versions of the ground-truth frame $x_i^0$ and the generated sample $\hat{x}_i^0 := f_\theta(x_i^t, t)$, respectively. The full training objective combines the sCM loss with adversarial supervision:

$$\mathcal{L} = \mathcal{L}_{\text{sCM}} + \lambda \mathcal{L}_{\text{adv}}. \tag{7}$$

**Sampling.** We apply a 4-step sampling where we select timesteps linearly across the range from $t_{\min}' = 0$ to $t_{\max}' = \frac{\pi}{2}$. For each step, we denoise the sample and inject noise to it corresponding to the next timestep, following Consistency Models (Song et al., 2023).

## 4.2 Speculative Sampling

Autoregressive models for video generation typically suffer from inference inefficiencies due to their memory-bound nature (Leviathan et al., 2023). To overcome this limitation, we introduce a speculative sampling technique designed to accelerate inference by enabling parallel prediction of multiple future frames. This method is grounded in the empirical observation that action sequences in interactive environments—such as gameplay scenarios—often exhibit short-term consistency. For instance, a player may continue performing the same action (e.g., walking or mining) over several consecutive frames. Leveraging this temporal redundancy, we propose to replicate the current action input $N$ times and feed these repeated inputs into the model in a single forward pass, allowing it to generate $N$ future frames speculatively.

After this speculative generation, we compare the predicted actions with the actual subsequent action inputs in the sequence. Once a discrepancy between the predicted and true actions is detected, all subsequent speculative frames beyond that point are discarded, and generation resumes from the last verified frame. This speculative approach significantly reduces the number of sequential decoding steps required during inference, thereby improving computational efficiency without sacrificing model accuracy or responsiveness.

## 4.3 Alleviating Error Accumulation with Noise Injection

The gap between training and autoregressive generation leads to error accumulation, resulting in quality degradation for subsequent frames. To mitigate accumulated error, inspired by previous works (Chen et al., 2024a; Valevski et al., 2024), we perturb the context frames by adding a small amount of Gaussian noise to the previously generated frames during sampling. This noise injection discourages the model from overly relying on past outputs by signaling that the context frames may be imperfect, thereby mitigating error accumulation and promoting more robust generation.

## 5 Experiments

We evaluate NFD on a large-scale action-conditioned video generation task, which consists of paired data comprising recorded gameplay videos and their corresponding action sequences.

**Dataset and Preprocessing.** We utilize the VPT dataset (Baker et al., 2022) for training and evaluation. Following MineWorld (Guo et al., 2025), to reduce noise and ambiguity during model training, we exclude frames that lack recorded actions as well as those captured when the graphical user interface (GUI) is open. The filtered data is randomly partitioned into training, validation, and test sets, comprising approximately 10M, 0.5K and 1K video clips, respectively. For both training and evaluation, each video frame is resized to a resolution of $384 \times 224$, which preserves the original aspect ratio while maintaining sufficient visual experience. We use 32 context frames during training, and evaluation on 16 frames to align with previous work (Guo et al., 2025).

**Implementation Details.** To enable frame-level interaction with the model, we employ a 2D variational autoencoder (Li et al., 2024) to tokenize each frame into continuous tokens. The tokenizer gives $16\times$ spatial compression and transforms each frame into $24 \times 14$ tokens. To improve recstruction quality, we fine-tune the decoder of the pre-trained tokenizer on our training data following previous practice (Tang et al., 2024). For the NFD base model training, we use the Adam optimizer (Kingma & Ba, 2015) with a learning rate of $1e$-4. For the consistency distillation, We use a two-stage strategy proposed by SANA-Sprint Chen et al. (2025). The first stage involves fine-tuning the pre-trained NFD for 100K steps at a learning rate of $1e$-4, and then we perform distillation, where we apply learning rates of $2e$-6 by default. All training is conducted on AMD MI300X GPUs with PyTorch (Paszke et al., 2019).

**Baselines.** We compare our method against the discrete autoregressive approach introduced in MineWorld (Guo et al., 2025), which serves as a strong baseline for assessing visual quality and sampling efficiency. We also add Oasis (Decart et al., 2024), an open-sourced diffusion-based world model on Minecraft, as a baseline, where we train a model using their model architecture and the same data as NFD (denoted as Oasis*). We further distill NFD with the method proposed in CausVid (Yin et al., 2024c), with their open-sourced codebase (denoted as CausVid*).

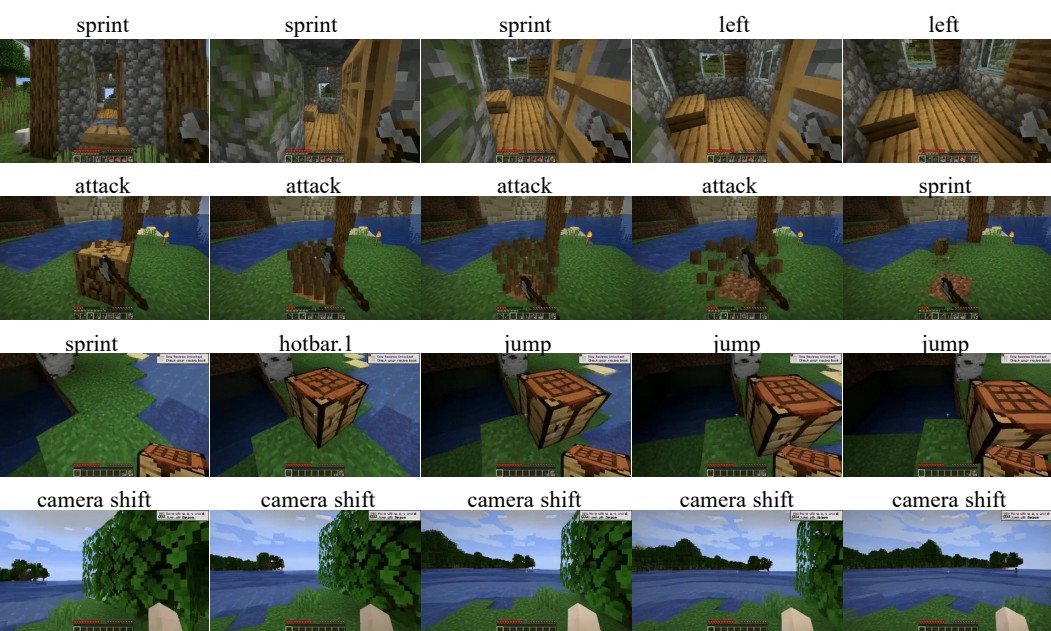

Figure 2: Qualitative results of the generated videos. Each row depicts a sequence of frames generated in response to a specific action command, such as sprint, attack, jump, and camera shift.

Table 1: Quantitative comparison with the baselines on both sampling efficiency and visual quality. NFD denotes the model trained using a Flow Matching objective and sampled with DPM-Solver++ using 18 sampling steps, while NFD+ indicates the accelerated variant of the model, incorporating consistency distillation and speculative sampling, and employing only 4 sampling steps. Oasis* refers to Oasis trained on the same data as ours. CausVid* refers to distilling NFD using the method proposed in CausVid. All FPS are evaluated on a NVIDIA A100 GPU with batch size of 1.

| Method | Param. | FPS ↑ | FVD ↓ | PSNR ↑ | LPIPS ↓ | SSIM ↑ |
|---|---|---|---|---|---|---|
| Oasis (Decart et al., 2024) | 500M | 2.58 | 377 | 14.38 | 0.53 | 0.36 |
| Oasis* | 500M | 2.58 | 213 | 16.73 | 0.38 | 0.43 |
| CausVid* (Yin et al., 2024c) | 130M | 6.15 | 363 | 14.83 | 0.46 | 0.37 |
| | 300M | 5.91 | 246 | 15.13 | 0.45 | 0.38 |
| MineWorld (Guo et al., 2025) | 700M | 3.18 | 231 | 15.32 | 0.44 | 0.38 |
| | 1.2B | 3.01 | 227 | 15.69 | 0.44 | 0.41 |
| | 130M | 7.51 | 220 | 16.34 | 0.40 | 0.43 |
| NFD | 310M | 6.15 | 212 | 16.46 | 0.38 | 0.44 |
| | 774M | 3.60 | **184** | **16.95** | **0.35** | **0.45** |
| | 130M | **42.46** | 246 | 16.50 | 0.38 | **0.44** |
| NFD+ | 310M | 31.14 | 227 | 16.83 | 0.35 | 0.43 |
| | 774M | 24.07 | **203** | **16.85** | **0.34** | **0.44** |

**Evaluation Metrics.** To evaluate visual fidelity, we adopt standard metrics including Fréchet Video Distance (FVD) (Unterthiner et al., 2018), Peak Signal-to-Noise Ratio (PSNR), Learned Perceptual Image Patch Similarity (LPIPS) (Zhang et al., 2018), and Structural Similarity Index Measure (SSIM) (Wang et al., 2004). To quantify sampling efficiency, we report the generation throughput in Frames Per Second (FPS) on a NVIDIA A100 GPU with batch size of 1.

Table 2: Ablation study on Speculative Sampling. We validate Speculative Sampling on different model size and different number of $N$. The results demonstrate that Speculative Sampling produces significant acceleration up to $1.26\times$.

| Param. | Method | Sampling Steps | FPS ↑ | Speed↑ |
|--------|--------|----------------|-------|--------|
| 130M | NFD+ w/o Speculative Sampling | 4 | 33.57 | 1.00× |
| | Speculative Sampling ($N$=2) | 4 | **42.46** | **1.26×** |
| | Speculative Sampling ($N$=3) | 4 | 40.76 | 1.21× |
| | Speculative Sampling ($N$=4) | 4 | 39.63 | 1.18× |
| 310M | NFD+ w/o Speculative Sampling | 4 | 26.15 | 1.00× |
| | Speculative Sampling ($N$=2) | 4 | 31.14 | 1.19× |
| | Speculative Sampling ($N$=3) | 4 | 31.22 | 1.19× |
| | Speculative Sampling ($N$=4) | 4 | **31.66** | **1.21×** |
| 774M | NFD+ w/o Speculative Sampling | 4 | 21.13 | 1.00× |
| | Speculative Sampling ($N$=2) | 4 | **24.07** | **1.14×** |
| | Speculative Sampling ($N$=3) | 4 | 23.64 | 1.12× |
| | Speculative Sampling ($N$=4) | 4 | 23.23 | 1.10× |

## 5.1 MAIN RESULTS

We present a comparative analysis of our proposed method against state-of-the-art baselines in Tab. 1, highlighting both sampling efficiency and visual quality of the generated videos. In Tab. 1 and the subsequent experiments, NFD refers to the model trained using a Flow Matching objective and sampled with DPM-Solver++ (Lu et al., 2022) using 18 function evaluations. NFD+ denotes the accelerated variant of the model, incorporating consistency distillation and speculative sampling, and employing only 4 sampling steps.

The results in Tab. 1 demonstrate that NFD consistently outperforms prior autoregressive models such as Oasis (Decart et al., 2024) and MineWorld (Guo et al., 2025) across multiple metrics. Specifically, NFD (310M) achieves a FVD of 212 and a PSNR of 16.46, outperforming MineWorld (1.2B) which has FVD of 227 and PSNR of 15.69, while running at 6.15 FPS, more than $2\times$ faster. NFD+ offers substantial speedups due to its efficient sampling strategy: the 130M and 310M models achieve 42.46 FPS and 31.14 FPS, respectively—surpassing all baselines by a large margin. Despite this acceleration, NFD+ maintains competitive visual quality, achieving a PSNR of 16.83 and FVD of 227 with 310M parameters, comparable to the best results among larger MineWorld models. We also find that applying CausVid to distill NFD yields significantly worse performance, consistent with their claim that CausVid is not well-suited for causal teachers.

We also provide qualitative results of the generated videos in Fig. 2, which showcase diverse action-conditioned sequences sampled by NFD. Each row depicts a sequence of frames generated in response to a specific action command, such as sprint, attack, jump, and camera shift. These qualitative results further substantiate that NFD not only achieves high quantitative performance but also excels in generating high-fidelity video sequences that are responsive to diverse action inputs.

## 5.2 ABLATION STUDIES

**Accelerating Inference via Speculative Sampling.** To support real-time interactive video generation, NFD integrates Speculative Sampling, a technique that enables the parallel generation of multiple future frames, thereby reducing latency during inference. As shown in Tab. 2, increasing the parallelism level to $N = 2, 3, 4$ consistently improves efficiency across both the 130M and 310M models. Notably, for the 130M model, setting $N = 2$ produces significant acceleration of $1.26\times$, and achieves the optimal trade-off between decoding parallelism and computational cost. Based on this observation, we adopt $N = 2$ as the default configuration for all models evaluated in subsequent experiments. These findings underscore the effectiveness of Speculative Sampling in enhancing the practicality of NFD for real-time applications.

Table 3: Ablation on sCM noise distribution. We empirically use (0.0,1.6) by default.

| Loss | $(P_{mean}, P_{std})$ | FVD ↓ | PSNR ↑ | LPIPS ↓ | SSIM ↑ |
|---|---|---|---|---|---|
| $L_{sCM} + L_{adv}$ | (0.0, 1.6) | **246** | 16.50 | **0.38** | **0.44** |
| $L_{sCM} + L_{adv}$ | (0.2, 1.6) | 269 | **16.54** | **0.38** | 0.42 |
| $L_{sCM}$ | (0.0, 1.6) | 266 | 16.13 | 0.40 | 0.42 |
| $L_{sCM}$ | (0.2, 1.6) | 285 | 15.66 | 0.42 | 0.40 |

**Ablation on sCM Noise Distribution.**   As described in Sec. 4.1, the noise schedule for sCM is defined as $t_i = \arctan\left(\frac{e^\tau}{\sigma_d}\right)$, where $\tau \sim \mathcal{N}(P_{\text{mean}}, P_{\text{std}}^2)$. We investigate the impact of the distribution of $\tau$, as summarized in Tab. 3. Our results indicate that the choice of $\tau$ distribution plays a critical role in performance—models trained with similar $P_{\text{mean}}$ values can exhibit noticeably different performance.

We further evalaute the effectiveness of adversarial loss by disabling it and using only the $L_{\text{sCM}}$ as the training objective. We observe a significant drop in generation quality given the same $\tau$ distribution, highlighting that the adversarial loss plays a critical role in enhancing the fidelity of generated videos.

**Ablation on Scalability.**   To evaluate the scalability of our approach, we train NFD models of varying sizes. As shown in Tab. 1, increasing the model size consistently leads to improved visual quality. Notably, the 774M-parameter NFD achieves an FVD of 184, establishing a new state-of-the-art among all NFD variants trained under the same paradigm.

**More Ablations.**   Additional experimental results are provided in the appendix, including ablations on KV cache usage, alternative model architectures, action conditioning, and different distillation strategies, etc.

## 6 CONCLUSION

We introduced Next-Frame Diffusion (NFD), a novel diffusion-based video generation framework designed to combine the high-fidelity synthesis capabilities of diffusion models with the temporal causality and controllability of autoregressive approaches. By incorporating block-wise causal attention, NFD enables parallel token sampling within individual frames while preserving strict autoregressive dependencies across frames. To address the challenges of real-time inference, we further proposed several innovations that significantly enhance sampling efficiency and visual quality: fast sampling via video-domain consistency distillation, speculative sampling by leveraging parallelism. Experiments on a large-scale video generation benchmark demonstrate that NFD achieves autoregressive video generation at a rate exceeding 30 FPS, while maintaining high visual quality.

**Limitations.**   While NFD demonstrates strong performance in terms of both visual fidelity and sampling efficiency, several limitations remain. First, the current implementation of NFD has a limited temporal context window (i.e., 32 frames), which could be further extended to enable tasks requiring sustained coherence or planning over extended horizons. Second, NFD is trained exclusively on Minecraft gameplay data. Expanding to more diverse datasets could improve robustness and applicability across a broader range of environments. Third, NFD is trained and evaluated at a fixed resolution ($384 \times 224$), chosen to preserve aspect ratio and balance quality with computational efficiency.

**Future Works.**   Given the promising results of both visual quality and sampling efficiency, future work should continue to scaling NFD to larger models and higher resolutions. In addition, future work could explore pretraining or finetuning on a broader range of video datasets, including real-world environments.

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

# A  APPENDIX

## A  IMPLEMENTATION DETAILS

**Hyperparameters.**  Tab. 4 summarizes the hyperparameters used for pretraining the NFD models. For fine-tuning, as discussed in Sec.5, we adopt the same set of hyperparameters as used during pretraining. The hyperparameters for distilling NFD+ are detailed in Tab. 5.

Table 4: Hyperparameters used in NFD.

| NFD | |
| --- | --- |
| Learning Rate Scheduler | constant |
| Learning Rate | $1e^{-4}$ |
| Batch Size | 96 |
| Warmup Steps | 10000 |
| Max Norm | 1.0 |
| Optimizer | AdamW |

Table 5: Hyperparameters used in NFD+.

| NFD+ | |
| --- | --- |
| Learning Rate Scheduler | constant |
| Learning Rate | $2e^{-6}$ |
| Batch Size | 512 |
| Warmup Steps | 0 |
| Max Norm | 0.1 |
| Optimizer | AdamW |

**Model Configurations.**  To validate the scalability of our training paradigm, we trained NFDs of varying sizes. Specifically, as shown in Tab. 6, we tuned key architectural hyperparameters, including the hidden dimension, the MLP dimension, the number of attention heads, and the number of layers. This allowed us to explore the effects of model size on performance and ensure the scalability of our approach across different capacity regimes.

Table 6: The configuration of different size of models.

| | Hidden Dim. | MLP Dim. | Num. Heads | Num. Layers |
| --- | --- | --- | --- | --- |
| 130M | 768 | 2048 | 12 | 12 |
| 310M | 1024 | 2730 | 16 | 16 |
| 774M | 1536 | 4096 | 24 | 18 |

**Detailed Algorithm of Speculative Sampling.** We present a concise and efficient implementation of the Speculative Sampling strategy. The algorithm assumes that the action for the next `nframe` time steps remains unchanged from action $i$. Accordingly, the model conditions on a repeated action input and generates `nframe` speculative frames in parallel. After generation, we verify whether the generated frames align with their intended actions and retain only those that are correctly generated. The process is repeated until the entire video sequence is synthesized.

```python
class NFD:
    def generate_nframe(model, vid, act):
        """
        model: Distilled NFD+ model
        vid: Input video tensor
        act: Action sequence tensor
        """
        x = vid[:, :n_prompt_frames]
        scheduler.set_timesteps(num_steps, ...)
        i = n_prompt_frames
        while i < total_frames:
            chunk = noise[:, i:i+nframe]
            x = concat(x, chunk)
            for t in timesteps:
                x_ctx = add_context_noise(x[:, :i])
                context = act[:, :i+1]
                repeat_act = repeat(act[:, i:i+1], times=nframe-1)
                act_seq = concat(context, repeat_act)
                pred = model(x_ctx / sigma_data, t, act_seq)
                latents, denoised = scheduler.step(pred, ...)
                x[:, -nframe:] = denoised
            i += nframe if same_action(act[:, i:i+nframe]) else
                first_change_idx + 1
            x = x[:, :i]
        return x
```

## B  ADDITIONAL EXPERIMENTS

**Quantitative Results on VBench.** We utilize VBench (Zhang et al., 2024) to further evaluate the generative capabilities of our model. For our assessment, we focus on three key metrics: Subject Consistency (Subj. Cons.), Image Quality (Image Qual.), and Dynamic Degree (Dyna. Degree).

Table 7: Quantitative comparison with the baselines on VBench. Compared to MineWorld (Guo et al., 2025), our approach achieves competitve results in both VBench and FVD, while offers $10\times$ speedup.

| Method | Param. | FPS ↑ | FVD ↓ | Subj. Cons. ↑ | Image Qual. ↑ | Dyna. Degree ↑ |
|--------|--------|-------|-------|---------------|---------------|----------------|
| MineWorld | 700M | 3.18 | 231 | 0.859 | 0.673 | **1.000** |
| NFD+ | 310M | **31.14** | **227** | **0.861** | **0.684** | 0.995 |

**Accelerating Inference via KV Caching.** Standard KV cache commonly used for the iterative decoding process can lead to accumulation errors. To address this, we cache the KVs of the noisy

Table 8: Ablation study on KV Cache. Caching noisy features at the first denoising step gives a speedup of $1.33\times$.

| Param. | Method | Sampling Steps | FPS ↑ | Speed↑ |
|--------|--------|----------------|-------|--------|
| 774M | NFD+ w/o KV Cache | 4 | 15.92 | 1.00$\times$ |
| | NFD+ w/. KV Cache | 4 | **21.13** | **1.33**$\times$ |

versions of the generated frames. Specifically, since the timestep associated with previous frames remains constant throughout the decoding process, we compute and cache their KVs during the first denoising step and reuse them across all subsequent denoising steps.

In Tab. 8, we present a quantitative comparison of NFD+ with and without the KV Cache. The results focus on the 774M NFD+ model with 4 sampling steps. Importantly, enabling the KV Cache for the largest NFD+ configuration yields a speedup of $1.33\times$.

**Comparison to Previous Works.** To train a Video Diffusion Model, most prior works adopt a variant of DiT or U-Net as the backbone (Kim et al., 2025; Deng et al., 2024; Valevski et al., 2024; Chen et al., 2024a), with the key difference lying in how visual tokens attend to one another. As an ablation, we also evaluate a 2D+1D attention variant for comparison as mentioned in Tab. 9, which is adopted by Oasis (Decart et al., 2024). In particular, Block Causal attention achieves higher FPS with comparable visual quality while using fewer parameters, highlighting its parameter efficiency.

Table 9: Comparison of different Model Architecture.

| Method | Param. | FPS ↑ | FVD ↓ | PSNR ↑ | LPIPS ↓ | SSIM ↑ |
|--------|--------|-------|-------|--------|---------|--------|
| Block Causal attention | 310M | **6.15** | **212** | 16.46 | **0.38** | **0.44** |
| 2D+1D (Oasis*) | 500M | 2.58 | 213 | **16.73** | 0.38 | 0.43 |

**More Ablations.** We conducted several ablation studies to evaluate the contributions of different components in our framework. First, we compared our proposed method (NFD+ with sCM) against an alternative variant using CausVid. As shown in Tab. 10, NFD+ achieves superior video quality metrics, reinforcing the advantages of our chosen approach.

We further examined the effect of 3D Normalization by replacing it with the normalization strategy from the original sCM paper. As reported in Tab. 13, the removal of 3D Normalization leads to performance degradation across most metrics, demonstrating its importance. Finally, we analyzed the impact of fine-tuning the VAE decoder. Tab. 12 shows that fine-tuning yields significant gains in PSNR, SSIM, LPIPS, and rFID, confirming its effectiveness for improving reconstruction quality.

Beyond the pretrained models listed in Tab. 1, we further explore the impact of different action conditioning strategies by training multiple variants of NFD. As shown in Tab. 11, applying conditioning via AdaLN-Zero consistently leads to significant improvements in FVD, highlighting its effectiveness in guiding high-fidelity video generation.

Table 10: Comparison of methods on video quality metrics.

| Method | FVD ↓ | PSNR ↑ | LPIPS ↓ | SSIM ↑ |
|--------|-------|--------|---------|--------|
| NFD+ (NFD w/ sCM) | 246 | **16.85** | **0.38** | **0.44** |
| CausVid* | 363 | 14.83 | 0.46 | 0.37 |

## C    CASE STUDY

We have included additional video results. Specifically, we prompt both MineWorld 700M and NFD+ 310M using the same input frame and actions, allowing for a direct comparison of their outputs.

Table 11: Ablation study on action conditioning. We compare different conditioning strategies on the same model. adaLN-Zero outperforms other baselines which aligns with image DiT (Peebles & Xie, 2023).

| Conditioning | Params. | FVD ↓ | PSNR ↑ | LPIPS ↓ | SSIM ↑ |
|---|---|---|---|---|---|
| adaLN-Zero | 130M | **220** | 16.34 | 0.40 | 0.43 |
| cross-attention | 158M | 244 | **16.39** | 0.40 | **0.44** |
| in-context | 130M | 223 | 16.32 | **0.39** | **0.44** |

Table 12: Comparison of VAE models with and without finetuning.

| Method | PSNR ↑ | SSIM ↑ | LPIPS ↓ | rFID ↓ |
|---|---|---|---|---|
| VAE (w/. finetune) | **33.15** | **0.90** | **0.07** | **12.61** |
| VAE (w/o finetune) | 28.07 | 0.85 | 0.12 | 16.94 |

**Consistency Across Frames.**   While both models can generate visually clear outputs given previous frames and the current action, NFD+ demonstrates superior temporal consistency, particularly in long-context scenarios. As shown in Fig. 3, NFD+ preserves a stable and coherent ground even after a significant camera movement, whereas MineWorld introduces visible artifacts and distortions.

**Details Aligned with Physical Properties.**   NFD+ demonstrates a stronger ability to preserve fine-grained physical properties, even as objects undergo changes in position or shape. As shown in Fig. 4, during the door-opening sequence, NFD+ accurately captures the door's geometry, maintaining its shape and structural integrity. In contrast, MineWorld introduces an artificial line between the two doors and fails to retain detail in the right portion of the door, indicating limitations in modeling object-level consistency.

**Visual Memorization.**   We observe that NFD+ consistently reconstructs previously seen objects with high fidelity. As generation progresses shown in Fig. 5, MineWorld appears to forget the brown block, introducing distortions in its appearance. In contrast, NFD+ preserves the block's size, position, and structure, demonstrating stronger object-level memorization over time.

## D   USE OF LARGE LANGUAGE MODELS

In accordance with ICLR 2026's policies on Large Language Model (LLM) usage, we confirm that no LLMs were used at any stage of preparing this submission. All text, figures, and code were written and verified solely by the human authors.

## E   ETHICS STATEMENT

This work adheres to the ICLR Code of Ethics. Our research does not involve human subjects, sensitive personal data, or applications that raise immediate safety or fairness concerns. We do not foresee any direct negative societal impacts from the methods proposed. All experiments were conducted responsibly with publicly available datasets, respecting licensing terms and privacy considerations.

## F   REPRODUCIBILITY STATEMENT

We have taken several steps to ensure the reproducibility of our results. All details of the proposed method, including model architectures, training procedures, and hyperparameters, are described in the main text and appendix. We additionally provide pseudocode and descriptions of experimental setups. Datasets used in our experiments are publicly available. Source code will be released to facilitate replication of our findings.

Table 13: Comparison of NFD+ with and without 3D Normalization.

| Method | FVD ↓ | PSNR ↑ | LPIPS ↓ | SSIM ↑ |
|---|---|---|---|---|
| NFD+ (w/ 3D Normalization) | **246** | **16.85** | **0.38** | **0.44** |
| NFD+ (w/o 3D Normalization) | 276 | 16.35 | 0.42 | 0.40 |

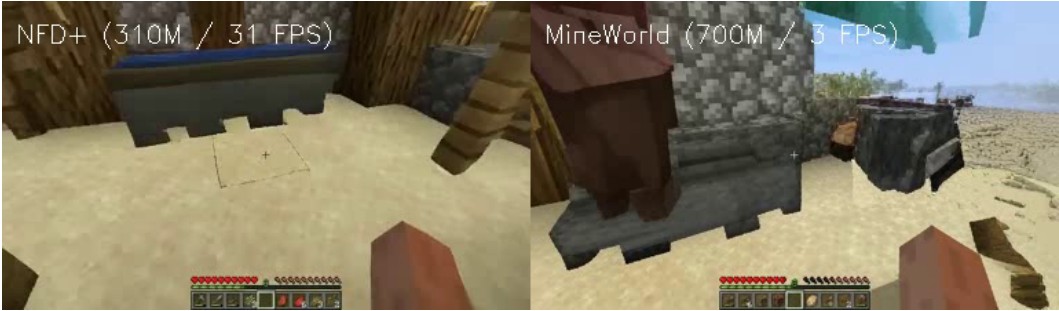

Figure 3: Frames generated by NFD+ and MineWorld respectively, which illustrates the superior temporal consistency achieved by NFD+. Despite a significant camera movement, NFD+ preserves a coherent and artifact-free background, whereas MineWorld introduces noticeable background distortions. This highlights NFD+'s robustness in maintaining scene integrity and temporal consistency.

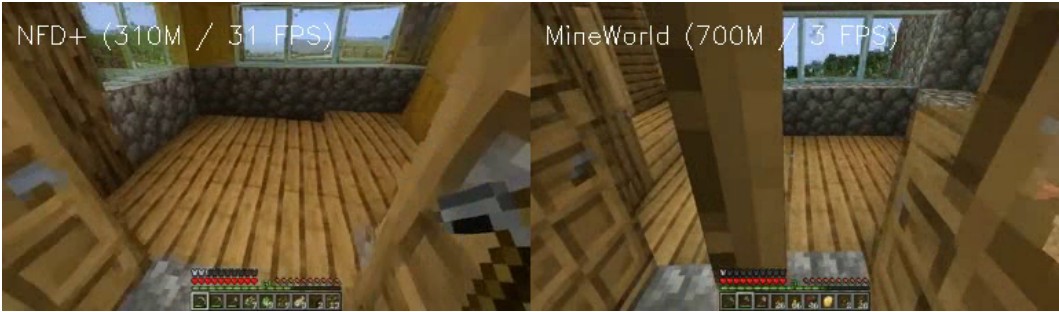

Figure 4: Frames generated by NFD+ and MineWorld respectively, which illustrates a door-opening sequence. NFD+ successfully renders the doors opening widely with no visible distortions, maintaining structural coherence. In contrast, MineWorld introduces a spurious artifact—a distorted line appearing between the two doors—highlighting its struggle with fine-grained object interactions.

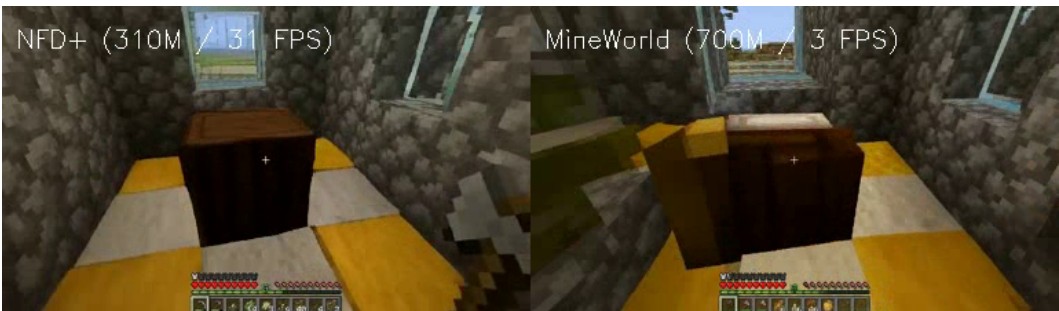

Figure 5: In this case, both models have previously encountered the brown block. NFD+ successfully reconstructs the block with high fidelity, while MineWorld fails to do so. This highlights the effectiveness of NFD+'s memorization capability in preserving object identity over time.

