# OpenReview forum: "Playing with Transformer at 30+ FPS via Next-Frame Diffusion"
_ICLR.cc/2026/Conference — ICLR 2026 Conference Withdrawn Submission_

### Official Review · Reviewer_mZoC · 2025-10-28

**Soundness:** 3
**Presentation:** 3
**Contribution:** 3
**Rating:** 6
**Confidence:** 4

**Summary:**

This paper proposes a pixel-level, next-frame video prediction diffusion model that is conditioned on actions and optimized for efficient inference. The authors employ an efficient sampling strategy and a few-step model distillation technique to accelerate the model. Furthermore, a multi-frame prediction method is introduced to enhance generation speed.

**Strengths:**

1. **Targeted Engineering for Acceleration:** The paper presents specific engineering designs aimed at accelerating video diffusion models, which is a significant contribution to the field.

2. **State-of-the-Art Performance:** The proposed model achieves state-of-the-art visual quality and sampling efficiency when compared to baseline methods, as demonstrated by the quantitative results.

**Weaknesses:**

1. **Lack of Qualitative Comparison:** Despite achieving strong numerical results, the paper lacks a qualitative comparison with baseline methods, such as a user study. Visual examples are crucial for a comprehensive evaluation of generative models.

2. **Hybrid Contribution:** The paper's contributions appear to be a mix of engineering and algorithmic improvements. However, the engineering enhancements are not explored to their full potential, and the algorithmic advancements are largely adaptations of existing work.

**Questions:**

1. **Source of Improvement:** Could you provide a more detailed analysis of the source of the model's improvements?

    - In Table 1, the performance of NFD-310M is comparable to Oasis-500M, despite having fewer parameters. A deeper analysis of the architectural design choices that lead to this efficiency would be beneficial.

2. **Necessity of a User Study:** Given the subjective nature of video quality assessment, a user study would significantly strengthen the paper's claims, especially for evaluating the model's performance on action sequences created by humans.

3. **Effectiveness of "Speculative Decoding":** The efficacy of the proposed "speculative decoding" method seems highly dependent on the frequency of action changes within the dataset.

    - What is the action change frequency in the test set?

    - How does the "speculative decoding" method perform when the action change frequency is high or erratic?

    - **Clarification on Terminology:** The term "speculative decoding" as used here appears to differ from its common usage in the Large Language Model (LLM) community, where a smaller model predicts and a larger model verifies. Have you considered a similar approach, for instance, using a one-step prediction and then verifying it based on mathematical conditions like temporal consistency or proximity to the subsequent step's result?

4. **Dynamic Frame Repetition in “speculative decoding“:** The NFD+ series models achieve real-time performance (24-42 FPS). Have you considered implementing a dynamic `N` (frame repetition number) that could adapt in real-time to user action inputs? This could allow for more interactive and responsive generation by modifying the repetition times of the current action context on the fly.

5. **Performance Discrepancy:** Could you explain the significant difference in inference speed between CausVid\* and NFD+? Given that both models have a similar number of parameters and, presumably, the same number of sampling steps (4 for the distilled models), the reason for this performance gap is not immediately clear.

---

### Official Review · Reviewer_5NKJ · 2025-10-30

**Soundness:** 3
**Presentation:** 3
**Contribution:** 2
**Rating:** 4
**Confidence:** 4

**Summary:**

This paper proposes a **next-frame diffusion** framework aimed at achieving real-time video generation. The work highlights two main contributions:

1. **Consistency distillation** is introduced into video diffusion models, reducing the number of diffusion sampling steps to only a few.
2. **Speculative sampling** is employed to predict multiple frames simultaneously, significantly accelerating inference.

Together, these methods enable real-time generation at up to **30 FPS**.

**Strengths:**

- The paper is **well-structured** and **clearly written**, making the main ideas easy to follow.
- The **methodology and motivation** are clearly articulated.
- The **experimental comparisons** are straightforward and provide solid empirical evidence for the proposed improvements.

**Weaknesses:**

- The **novelty** of the contributions is limited. Both *consistency distillation* and *speculative sampling* are existing ideas that have been explored in prior works.
- The paper primarily adapts these known techniques to the *next-frame diffusion* setting, resulting in a contribution that feels **incremental rather than groundbreaking**.
- The paper would benefit from a deeper analysis or theoretical insight to strengthen its originality.

**Questions:**

1. **FVD Evaluation**
   It is unclear at which autoregressive (AR) step the reported FVD results are measured.
   Since AR methods are prone to error accumulation, the authors should compare FVD across multiple AR step lengths to better understand how performance degrades over time.
   This is especially important in acceleration scenarios, where such effects can be amplified.

2. **Speed vs. AR Step Length**
   As the AR step length increases, the **cache size** grows correspondingly, which should impact generation speed.
   For long video sequences, per-frame latency likely increases significantly.
   Instead of reporting a single speed value, the paper should provide a **curve of generation speed versus AR step length** for a more complete performance picture.

3. **Speculative Sampling Quality Control**
   The paper mentions that speculative samples are rejected only when the predicted and actual actions differ, but it does not discuss how **frame quality** is considered.
   If the multi-frame predictions are of poor quality, are they still accepted?
   The current experiments focus on speed improvements but **omit comparisons of generation quality** under speculative sampling, which is essential for evaluating the trade-off between quality and speed.

---

### Official Review · Reviewer_5oQx · 2025-10-30

**Soundness:** 3
**Presentation:** 2
**Contribution:** 2
**Rating:** 4
**Confidence:** 4

**Summary:**

The paper proposes Next-Frame Diffusion (NFD), an autoregressive diffusion transformer for action-conditioned video generation. The key architectural idea is block-wise causal attention, which allows bidirectional attention within a frame while enforcing causality across frames so that all tokens of the next frame can be generated in parallel. To reach real-time speeds, the authors (i) extend continuous-time consistency distillation (sCM) from images to the video domain to cut diffusion steps to four, and (ii) introduce speculative sampling that pre-generates a few future frames under the assumption the current action persists, discarding any that disagree when the action actually changes. On Minecraft/VPT data, NFD attains >30 FPS on an A100 with a 310M model (NFD+ variant) at 384×224 resolution, and reports FVD/PSNR/LPIPS/SSIM competitive with or better than discrete autoregressive baselines like MineWorld, with a speed–quality trade-off between the undistilled NFD and the accelerated NFD+. The paper includes ablations for speculative sampling (up to 1.26× faster), sCM schedules, KV caching (up to 1.33× faster), and some architecture comparisons. Limitations acknowledge short temporal context (32 frames), Minecraft-only data, and fixed, relatively low resolution.

**Strengths:**

1. Clean combination of block-causal attention with diffusion-based frame-wise prediction, plus an explicit extension of consistency distillation to video. The speculative-sampling idea for action-conditioned video is straightforward but practical.
2. Results table transparently shows the speed–quality trade-off: e.g., NFD (310M) FVD 212 at 6.15 FPS vs NFD+ (310M) FVD 227 at 31.14 FPS.
3. The paper is easy to follow, with an architectural figure and explicit training/sampling equations, plus a pseudocode sketch for speculative sampling. Limitations are candid.

**Weaknesses:**

1. Novelty relative to prior “frame-wise” diffusion variants appears incremental. The work builds on a now-active line of next-frame diffusion with parallel per-frame token generation and noise-perturbed contexts (e.g., Diffusion Forcing, CausVid/self-forcing-style distillations). While the paper positions its sCM extension and speculative sampling as first, readers may see these as natural adaptations rather than conceptual leaps. A head-to-head with Diffusion Forcing and CausVid under identical data/tokenizers would better clarify novelty/margins beyond the included CausVid student result.
2. All results are Minecraft (VPT). The paper itself notes domain limitation. The qualitative scenes and the chosen 384×224 resolution are unlikely to stress long-range temporal coherence, complex motion, or photorealistic detail; hence the scalability of the approach to harder domains/resolutions remains unproven.
3. Speed claims hinge on specific settings and show a quality drop after distillation. The flagship 30+ FPS is reported for NFD+ (4 steps) with speculative sampling and KV caching at low resolution and batch size 1; undistilled NFD is ~6 FPS for 310M. Quality regresses (FVD 212->227) with NFD+, suggesting the distilled model pays a non-trivial perceptual cost for speed. More end-to-end latency breakdowns (tokenizer, I/O), memory bandwidth profiling, and multi-resolution scaling curves would strengthen the case.

**Questions:**

1. What are the measured FPS and quality when scaling to, say 480p or 720p, with 81/121 context frames? Any stability issues (e.g., error accumulation) and how do sCM and context-noise injections behave under those regimes?
2. Speculative sampling under non-stationary actions. How often are speculative frames discarded on validation logs? Please report (i) discard rate vs N, (ii) net wall-clock speedup vs. GPU utilization, and (iii) quality when actions change rapidly (camera whip-pans, frequent tool switches). A small user-study or scripted stress-test would help.
3. Ablations on block-causal attention vs. alternatives. The appendix compares “2D+1D” vs block-causal. Could you also show full 3D attention and windowed causal variants at matched compute to establish where the FPS gains primarily come from (attention pattern vs. implementation/kv-cache)?
4. Since NFD+ trades quality for speed, can you provide (a) user preference studies, (b) VBench breakdowns beyond the three metrics shown, and (c) samples at equal FPS budgets (e.g., NFD with fewer NFEs) to clarify whether sCM is the dominant factor?
5. Generalization beyond Minecraft. Any preliminary results on real-world or non-voxel games (e.g., driving/world-model datasets, Sekai) to test motion complexity and texture diversity? Even short proofs-of-concept would help address external validity concerns raised in your Limitations.

---

### Official Review · Reviewer_eMsJ · 2025-11-01

**Soundness:** 2
**Presentation:** 3
**Contribution:** 2
**Rating:** 4
**Confidence:** 4

**Summary:**

This paper introduces Next-Frame Diffusion (NFD), an autoregressive diffusion model designed for real-time video generation. NFD combines the strengths of diffusion models (high fidelity) and autoregressive models (temporal causality) to achieve over 30 FPS on an A100 GPU. Key innovations include block-wise causal attention, consistency distillation, and speculative sampling, which significantly enhance sampling efficiency and visual quality.

**Strengths:**

- Well-writing: The paper is well-written, clear, and concise, effectively communicating its ideas without unnecessary complexity.
- Compelling scalability analysis: The model's performance improves with increased size, indicating potential for further gains with larger models and datasets.

**Weaknesses:**

- **Concerns on incremental novelty.** The proposed method appears to integrate existing techniques such as consistency distillation and speculative decoding, which may limit its conceptual novelty and suggest an incremental contribution.
- **Limited data validation and strong assumptions in speculative decoding.**  The assumption that actions remain identical during speculative decoding may be overly restrictive; I expect evaluation on more complex datasets to better validate the robustness and acceleration efficiency of this approach.

**Questions:**

- See weaknesses.
- In short, the work presents Next-Frame Diffusion (NFD), an autoregressive diffusion model designed for real-time video generation.
- My main concerns are the novelty of this work, as well as the limited data validation and the strong assumptions in speculative decoding.

---

### Note · Authors · 2025-11-26

I have read and agree with the venue's withdrawal policy on behalf of myself and my co-authors.